# Scaling laws of shrinkage induced fragmentation phenomena

Roland Szatmári[1], Akio Nakahara[2], So Kitsunezaki[3] and Ferenc Kun[4,5⋆]

**1** Department of Experimental Physics, Doctoral School of Physics,
Faculty of Science and Technology, University of Debrecen,
P.O. Box 400, H-4002 Debrecen, Hungary
**2** Laboratory of Physics, College of Science and Technology, Nihon University,
7-24-1 Narashinodai, Funabashi, 274-8501, Japan
**3** Research Group of Physics, Division of Natural Sciences,
Nara Women's University, Nara, 630-8506, Japan
**4** Department of Theoretical Physics, Faculty of Science and Technology,
University of Debrecen, P.O. Box 400, H-4002 Debrecen, Hungary
**5** HUN-REN Institute of Nuclear Research (HUN-REN Atomki),
Poroszlay út 6/c, H-4026 Debrecen, Hungary

⋆ ferenc.kun@science.unideb.hu

## Abstract

We investigate the shrinkage induced breakup of thin layers of heterogeneous materials attached to a substrate, a ubiquitous natural phenomenon with a wide range of potential applications. Focusing on the evolution of the fragment ensemble, we demonstrate that the system has two distinct phases: damage phase, where the layer is cracked, however, a dominant piece persists retaining the structural integrity of the layer, and a fragmentation phase, where the layer disintegrates into numerous small pieces. Based on finite size scaling we show that the transition between the two phases occurs at a critical damage analogous to continuous phase transitions. At the critical point a fully connected crack network emerges whose structure is controlled by the strength of adhesion to the substrate. In the strong adhesion limit, damage arises from random microcrack nucleation, resembling bond percolation, while weak adhesion facilitates stress concentration and the growth of cracks to large extensions. The critical exponents of the damage to fragmentation transition agree to a reasonable accuracy with those of two-dimensional bond percolation. Our findings provide a novel insights into the mechanism of shrinkage-induced cracking revealing generic scaling laws of the phenomenon.

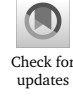

# 1 Introduction

Solid bodies break apart under the action of an external load, when the load exceeds the material's strength [1,2]. However, the process of breakup and the final outcome strongly depends on the way how the load is applied [3,4]. Dynamic fragmentation occurs when a large amount of energy is imparted to a solid within a short time e.g. by means of an explosion, projectile shooting, or free fall impact to a rigid wall. As a result, the solid rapidly disintegrates into a large number of pieces. Experiments have revealed that for a broad class of heterogeneous materials the mass (size) distribution of fragments follows a power law functional form with a high degree of robustness of the exponents [5–9]. Theoretical investigations uncovered that the origin of universality maybe a self-organized critical behaviour of cracking [8], a second order phase transition to the fragmented state [10–12], and the branching-merging mechanism accompanying the growth of dynamic cracks [13]. Recent experiments by Kooij et al. [14] further demonstrated that brittle fragmentation may yield either exponential or power-law fragment size distributions, depending on the driving conditions. This highlights that fragmentation can occur in distinct statistical regimes, a possibility we also investigate in the present work.

In contrast, gradual or slow fragmentation is often driven by internal stimuli, such as desiccation, cooling, or chemical processes, leading to volume changes and stress accumulation within the material [15–21]. Shrinkage-induced cracking is responsible for iconic natural crack patterns, including drying lake beds, polygonal permafrost formations on Earth and Mars, and columnar joints in volcanic lava flows [22,23]. Recently, the ability to control crack paths and pattern structures in shrinking layers has been demonstrated, offering promising industrial applications in fields such as surface design and microelectronic manufacturing [24–26].

Extensive experimental and theoretical studies of the last decades clarified the physics of crack nucleation and propagation in shrinking soft matter [21,22], and revealed the intricate laws of final-state fragmentation mosaics [27]. However, understanding the scaling laws governing the evolution of the crack ensemble in slow breakup phenomena and the emerging fragments still remained a fundamental challenge.

Here, we address this problem through a theoretical investigation of fragment evolution in a shrinking material layer. Using discrete element simulations, we show that the cracking process evolves through two distinct phases: an initial damaged phase, where cracks nucleate and grow but a dominant fragment maintains the layer's structural integrity; and a subsequent fragmentation phase, where the system disintegrates into numerous smaller fragments. The transition between these phases occurs at a well-defined critical damage threshold, corresponding to the formation of a fully connected crack network spanning the system. Most notably, finite-size scaling analysis reveals that the damage to fragmentation transition is analogous to continuous phase transitions. Remarkably, while adhesion strength influences the degree of heterogeneity of the stress field in the cracked layer, it has only a weak effect on

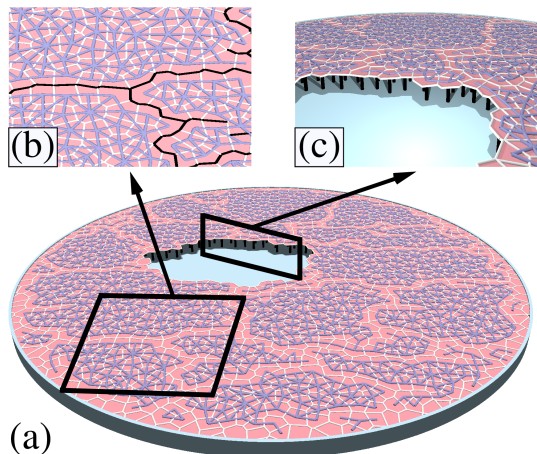

Figure 1: Construction of the discrete element model of restrained shrinking of a thin brittle layer. ($a$) The layer is discretized using space filling convex polygons obtained by a Voronoi tessellation with controlled disorder. ($b$) Polygons are connected by beams which exert cohesive forces and are able to break. Successive breaking of neighboring beams results in cracks in the layer. ($c$) Adhesion is represented by springs connecting the center of polygons to the substrate.

the critical exponents. The extracted critical exponents are found to be close to those of the two-dimensional bond percolation model [28], hinting at a possible universality connection. However, given the limited system sizes and uncertainties in scaling collapse, other universality classes cannot be ruled out. Below the critical point, the fragment mass distribution develops a power-law regime for small fragments and a distinct hump for larger ones, separated by a characteristic gap. Above the critical point, fragments undergo a binary splitting process, which drives the mass distribution to a log-normal form. These findings not only deepen our understanding of shrinkage-induced cracking but also have broad implications for natural systems and industrial applications, from desiccation cracks in soils to surface patterning in engineering applications.

## 2 Modeling shrinkage induced cracking

Following the pioneering works of Meakin [29] and Colina et al. [30], we have recently developed a discrete element model for the shrinkage-induced cracking of a thin material layer attached to a rigid substrate, which captures the key mechanisms of the cracking process. Here we briefly summarize the main steps of the model construction. For more details of the model and for the calibration of its parameters see Ref. [31].

The material layer is discretized by means of randomly shaped convex polygons which are obtained by a regularized Voronoi tessellation of a square [32]. The algorithm begins by overlaying a regular square lattice on the sample. For each plaquette of this lattice, a base point of the Voronoi construction is chosen at random within a smaller square centered on the plaquette. This procedure limits the number of polygon corners and allows the degree of structural disorder in the tessellation to be systematically controlled by varying the ratio of the smaller-square side length to the lattice spacing [32]. In order to avoid the disturbing effect of corners, from the initial square we cut out a circular sample of radius $R$. Polygons represent mesoscopic material elements which have three degrees of freedom in two dimensions, i.e. the two coordinates of the center of mass and a rotation angle. To form a solid body, the center of

mass of polygons which are adjacent in the initial tessellation are connected by beams of Young modulus $E$, which exert forces and torques as the polygons get displaced. The geometrical properties of beams are determined from the initial tessellation in such a way that the initial value of the natural length of a beam $l_0^{ij}$ connecting polygons $i$ and $j$, and its cross section $S_0^{ij}$ are obtained as the distance of the centers of mass and of the length of the common edges of the two polygons, respectively. Figure 1 illustrates the model construction.

To capture shrinking of the layer, the natural length of beams $l^{ij}$ is gradually decreased with time $t$ at a constant rate $s$ as $l^{ij}(t) = l_0^{ij}(1 - st)$, which results in a linearly increasing shrinkage strain $\varepsilon_s = st$. Forces and torques emerging in beams as a consequences of shrinking can be expressed in terms of the displacements and rotation angles of connected polygons (for details see Refs. [31, 33]). However, due to the isotropic shrinking of the disordered homogeneous system, the dominating component of the beam force is always the longitudinal one $F_n^{ij}$ which can be cast into the form

$$F_n^{ij} = D_b^{ij}\left(\left|\vec{r}^i - \vec{r}^j\right| - l^{ij}\right). \tag{1}$$

Here $\vec{r}^i$ and $\vec{r}^j$ are the position vectors of the two polygons connected by the beam, and $D_b^{ij}$ is the beam stiffness which depends on the Young modulus $E$ of the material and on the geometrical properties of beams in the initial lattice $D_b^{ij} = E S_0^{ij}/l_0^{ij}$. Adhesion is represented by connecting the center of polygons in their initial position $\vec{r}_0^i$ to the substrate by an elastic spring of stiffness $D_s$. As the polygons get displaced, due to the shrinking of beams, to the position $\vec{r}^i$ a restoring force emerges

$$\vec{F}_s = -D_s(\vec{r}^i - \vec{r}_0^i), \tag{2}$$

which makes shrinking constrained.

As the sample shrinks beams get overstressed and break according to a physical breaking rule

$$\left(\frac{\varepsilon_b^{ij}}{\varepsilon_{th}}\right)^2 + \frac{\max\left(|\Theta_i|, |\Theta_j|\right)}{\Theta_{th}} \geq 1, \tag{3}$$

where $\varepsilon_b^{ij}$ is the longitudinal strain of the beam, while $\Theta_i$ and $\Theta_j$ are the bending angles of the two beam ends. The breaking criterion takes into account that stretching and bending (shear) contribute to breaking so that the parameters $\varepsilon_{th}$ and $\Theta_{th}$ control the relative importance of the two breaking modes. It is an important feature of our model that the discretization of the sample on a random lattice of convex polygons is the only source of disorder so that the value of the Young modulus $E$, the spring stiffness $D_s$, and the breaking parameters $\varepsilon_{th}$, and $\Theta_{th}$ have fixed values for the entire system. For the parameter values used in the simulations see Ref. [31]. As beams break, micro-cracks nucleate along the common side of neighboring polygons and sequences of adjacent broken beams generate extended cracks in the sample (see Fig. 1(a, c)).

To generate the time evolution of the shrinking system we solve the equation of motion of polygons using a $5th$ order Predictor-Corrector scheme [34]. The breaking criterion Eq. (3) is evaluated in each iteration step and those beams which fulfill the condition are removed from the simulations. To ensure numerical stability, a velocity dependent damping force is applied. When two polygons which are not connected by beam come into contact, we let them overlap and introduce an elastic restoring force proportional to the overlap area. This repulsive force prevents crack faces to penetrate each other.

Simulations were performed for four different system size $R = 30, 60, 90, 120$, where length is measured in units of the average polygon size $l_0$. An important parameter of our study is the stiffness of the springs $D_s$ connecting the polygons to the substrate. To characterize the strength of adhesion, we introduce the dimensionless parameter $D_s/D_b$, where $D_b$ denotes the average value of the stiffness of beams $D_b^{ij}$ ($i, j = 1, \ldots, N$). Simulations were

performed varying $D_s$ in a broad range at a fixed value of the beam Young modulus $E$ so that the ratio $D_s/D_b$ covered the range $10^{-3} \leq D_s/D_b \leq 5$. At a given parameter set, averages were calculated over $K = 100$ simulations with different realizations of the structural disorder of the layer.

# 3 The effect of adhesion strength on the stress field around cracks

As a consequence of shrinking, beams of the layer get overstressed and break leading to the nucleation of cracks, which may grow further. Initially, crack nucleation occurs at the weakest spots of the layer, however, the dynamics of the growth of cracks and the gradual nucleation of new ones are strongly affected by the stress field emerging in the cracked layer. The strength of adhesion of the layer material to the underlying rigid substrate plays an important role in shaping the stress field around cracks. In the limit of strong adhesion $D_s/D_b \gg 1$, when a micro-crack nucleates, polygons can hardly move which prevents the release of stress around cracks. In the opposite limit of weak adhesion $D_s/D_b \ll 1$, polygons can undergo large displacements, which facilitates stress release over a larger area. To obtain a deeper insight into the effect of adhesion strength on the characteristic length scale of stress release, we performed numerical simulations. In an initial sample with a radius of $R = 120$, a crack of half-length $l_c = 50$ was introduced at the center along a diagonal. The layer was then subjected to slow shrinkage while preventing beam breakage. Simulations were halted at a strain of $st = 0.007$, at which point we analyzed the spatial distribution of the longitudinal component $F_n^{ij}$ of the beam force ahead of the crack. The spatial dependence of $F_n^{ij}$ is illustrated in the inset of Fig. 2(a), where beams are colored according to the value of the force component.

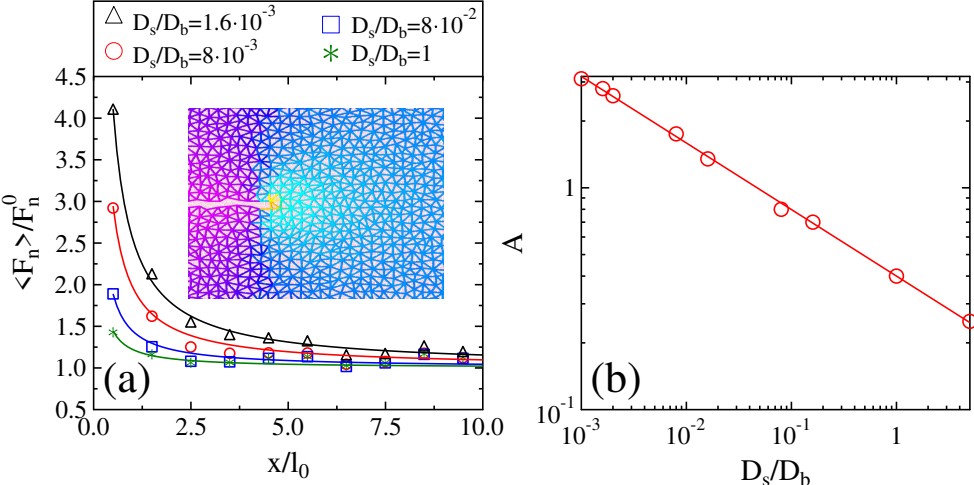

Figure 2: (a) Average value $\langle F_n \rangle$ of the longitudinal force of beams ahead of a crack as a function of the distance $x$ measured from the crack tip. Results are presented for four different values of the adhesion strength $D_s/D_b$. Along the vertical and horizontal axis the data are scaled with the value of the background force $F_n^0$ and of the average polygon size $l_0$, respectively. The continuous lines represent fits with Eq. (4). The spatial distribution of the longitudinal force is illustrated by the inset where beams are colored according to the value of $F_n^{ij}$. (b) The parameter $A$ controlling the range of load redistribution around crack tips as a function of the adhesion strength $D_s/D_b$. The straight line represents a power law of exponent $\kappa = 0.3$.

For the quantitative characterization, we determined the average value $\langle F_n \rangle$ of the longitudinal force component $F_n^{ij}$ of those beams which intersect the line of the crack. The average force $\langle F_n \rangle$ normalized by the background value $F_n^0$ is presented in Fig. 2($a$) as a function of the distance $x$ measured from the crack tip at four different values of the adhesion strength. It can be observed that with increasing $x$ the average force gradually decreases to the background value $\langle F_n \rangle / F_n^0 \approx 1$, however, the decay rate depends on the adhesion strength $D_s/D_b$. Our data analysis revealed that the force-distance curves can be described by the functional form

$$\frac{\langle F_n \rangle}{F_n^0} = Ax^{-\eta} + B\,, \tag{4}$$

where the additive constant is $B = 1$ for all the cases. Best fit was obtained using the same value of the exponent $\eta = 1$, however, the multiplication factor $A$ decreases with the adhesion strength $D_s/D_b$. Strong adhesion prevents stress relaxation; therefore, the force distribution remains nearly homogeneous even ahead of the crack tip, which is indicated by the low value of $A = 0.25$ obtained for $D_s/D_b = 5$. However, at lower $D_s/D_b$, the stress relaxation extends to large distances, and results in a high stress concentration ahead of the crack tip, which is captured by the significantly higher value of the parameter $A = 3.1$ at $D_s/D_b = 10^{-3}$. To quantify the degree of inhomogeneity of the stress field around cracks, we analyzed how the multiplication factor $A$ depends on $D_s/D_b$. A power law decay is evidenced in Fig. 2($b$)

$$A \sim (D_s/D_b)^{-\kappa}\,, \tag{5}$$

where best fit was obtained with the exponent $\kappa = 0.30 \pm 0.04$. Our numerical results show reasonable agreement with the analytical predictions of Ref. [35], which reported a power-law decay followed by an exponential cutoff. In our study, however, the limited resolution of the discrete random lattice and the relatively small system size prevented us from resolving the exponential cutoff. Nevertheless, the short-range nature of load redistribution at high adhesion is reflected in the low value of the multiplication factor $A$.

## 4 Structural evolution of shrinking induced crack patterns

Our simulations revealed that the adhesion strength $D_s/D_b$ has a strong effect on the structure of the evolving crack pattern. As the layer shrinks, the beams gradually become overstretched and break according to the condition Eq. (3). Despite the constant breaking parameters $\varepsilon_{th}$ and $\Theta_{th}$ the structural disorder imprinted in the polygonal lattice causes a local variation of the strength of the beams so that the cracks first nucleate in the weakest locations, i.e. at the longest and thinnest beams. Due to the high degree of isotropy and homogeneity of the system, cracks grow in random directions along with new nucleations in intact regions of the sample which can be observed in Fig. 3($a$) for the parameter value $D_s/D_b = 0.08$. Figure 3($b$) shows that as the system evolves a notable point emerges in the dynamics, where the growing cracks merge and form a connected crack network, along which the layer is broken down into many fragments. Fragments are completely surrounded by cracks, making them independent of each other. It has the consequence that further shrinking accumulates stress inside fragments which in turn results in crack nucleation and breaking of fragments typically into two further pieces. Once the connected crack network is formed the further evolution of the system is controlled by this binary breakup process (Fig. 3($c, d$)).

Since the adhesion strength $D_s/D_b$ controls the degree of inhomogeneity of the stress field in the cracking layer, it can have a strong effect both on the temporal evolution of the cracking and on the spatial structure of the resulting crack network. Strong adhesion (high $D_s/D_b$) gives rise to short cracks which can hardly advance due to the short range stress redistribution

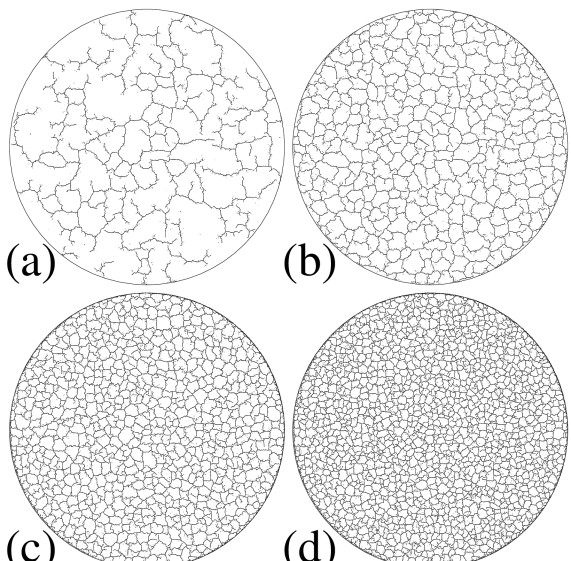

Figure 3: Time evolution of shrinking induced cracking of a circular layer of radius $R = 120$ at the adhesion strength $D_s/D_b = 8 \cdot 10^{-2}$. First, cracks nucleate at the weakest spots and grow in random directions $(a)$. Growing cracks merge and form a connected network along which the sample falls apart into fragments $(b)$. Further shrinking results in crack nucleation inside fragments which breaks them into two daughter pieces $(c, d)$.

in the layer and low stress concentration at the crack tips. Hence, in this parameter regime the cracking process is controlled by the structural disorder of the material. In case of weak adhesion (low $D_s/D_b$), the high stress concentration at the crack tip promotes crack growth so that after nucleation cracks typically grow and extend to longer length. To demonstrate how this mechanism affects the structure of the emerging crack network, in Fig. 4 we present snapshots of the layer right after the connected crack network is formed for several $D_s/D_b$ values. It can be observed that as the adhesion strength decreases from Fig. 4$(a)$ to $(d)$ less-and-less beam breakings are sufficient to create the connected crack network, since cracks are getting longer, and they form larger-and-larger fragments in the layer. Note that the crack pattern always has the isotropic cellular structure, similar to the experimental findings on the desiccation induced cracking of e.g. dense pastes [18, 24].

To quantify the degree of degradation of the shrinking layer, we introduce the cumulative damage $d(t)$ as the fraction of broken beams $d = N_b(t)/N_B$, where $N_B$ denotes the total number of beams in the initial sample, and $N_b(t)$ is the number of beams that break up to time $t$. To monitor the size reduction achieved during the cracking process, we determined the average mass of the fragments $\langle M_{av} \rangle$ as a function of damage $d$ for various adhesion strengths. The value of $M_{av}$ is first calculated from single samples as the ratio of the second and first moments of fragment masses

$$M_{av} = \frac{M_2}{M_1} \,, \tag{6}$$

where $M_q = \sum_i' m_i^q$ is the $q$th moment of fragment masses. The $\prime$ indicates that the largest fragment is omitted in the summation. Eventually, the average fragment mass $\langle M_{av} \rangle$ is obtained by averaging $M_{av}$ over a large number of samples. Figure 5$(a)$ shows that for each $D_s/D_b$ the $\langle M_{av} \rangle$ curves have a relatively sharp maximum at a well defined critical damage $d_c$. Since the largest fragment is omitted, the maximum marks the point of the dynamics, where the dominating large piece ceases to exist, i.e. where the connected crack network emerges

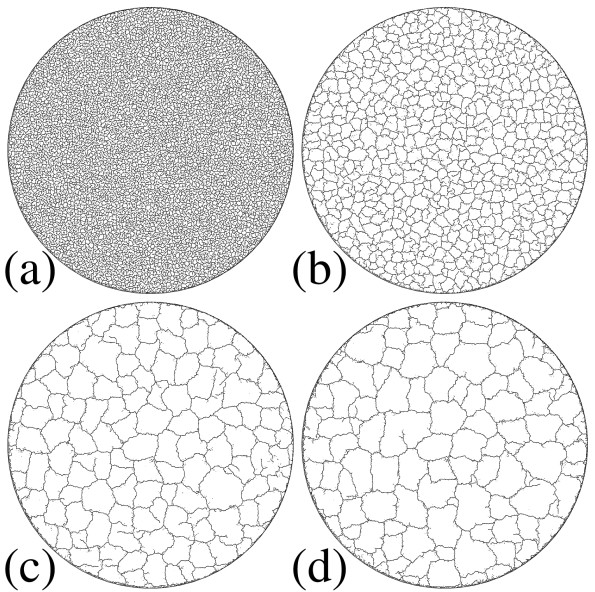

Figure 4: Snapshots of the cracking layer right after the connected crack network is formed at four different values of the adhesion strength $D_s/D_b$: (a) 1, (b) $8 \cdot 10^{-2}$, (c) $8 \cdot 10^{-3}$, (d) $1.6 \cdot 10^{-3}$.

and the layer falls apart into a large number of fragments. As the adhesion strength $D_s/D_b$ increases, the maximum shifts to higher damage values, which shows that a larger and larger number of micro-cracks must occur to form a spanning network, while the decreasing hight of the curves indicates the reduction of the fragment size.

To obtain a better overview of the evolution of the curves with the adhesion strength, in Fig. 5(b) the critical damage $d_c$ is presented as a function of $D_s/D_b$ for three system sizes $R$. It can be seen that $d_c$ monotonically increases for all $R$ values indicating that at stronger adhesion a higher amount of damage has to accumulate to form a connected crack network. As the adhesion strength increases, the stress field becomes more and more homogeneous in the cracked layer, and no stress concentrations can build up around crack tips. As a consequence, in the high $D_s/D_b$ limit, the cracking of the layer is dominated by random nucleation of micro-cracks governed by the structural disorder of the layer material. Hence, the spatial structure of the crack pattern of the shrinking layer becomes analogous to a bond percolation lattice, where bonds of the lattice, i.e. the edges of the original Voronoi mosaic are randomly occupied [28]. This bond percolation limit determines the upper bound of $d_c$ achieved for high $D_s/D_b$. In the case of low adhesion, the value of $d_c$ also approaches an asymptotic value for each system size $R$. As $D_s/D_b$ decreases, cracks can extend to higher length without being stopped by another propagating crack. Hence, the low adhesion limit of the critical damage $d_c$ is determined by the size of the system $R$, i.e. the $d_c$ curves level off when the path a crack can cover before being interrupted becomes comparable to $R$.

## 5 Transition from damage to fragmentation

Understanding how the shrinkage-induced breakup of the layer occurs as a result of gradual damage accumulation is a key question. Our analysis has revealed that the critical damage $d_c$ where the connected crack network emerges separates two qualitatively different regimes of the system. For damage levels below the critical point $d < d_c$, the crucial feature is that the layer retains a dominant fragment, which is significantly larger than all other fragments.

SciPost Phys. **19**, 142 (2025)

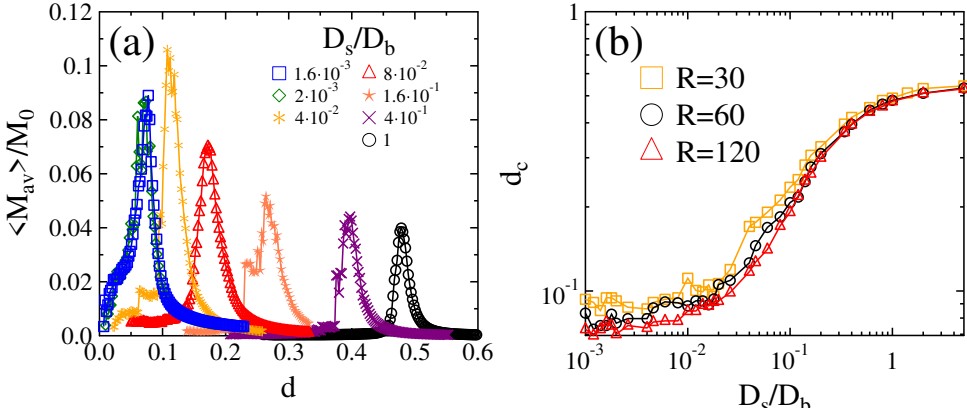

Figure 5: ($a$) The average fragment mass $\langle M_{av} \rangle$ normalized by the total system mass $M_0$ as a function of the cumulative damage $d$ for different adhesion strengths $D_s/D_b$. The sharp maximum marks the critical damage $d_c$, where the crack network becomes fully connected, leading to the breakup of the layer into a large number of fragments. ($b$) The critical damage $d_c$ as a function of the adhesion strength $D_s/D_b$ for three different system sizes $R$.

This behavior is illustrated in Fig. 6, where the average mass of the largest fragment $\langle M_{max} \rangle$ is compared to that of the second-largest one $\langle M_{max}^{2nd} \rangle$. In the subcritical regime $d < d_c$, it is evident that even the second largest fragment is significantly smaller than the largest one. As the shrinkage progresses, crack nucleation and growth lead to a gradual decrease in the size of the largest fragment, while the second-largest fragment grows and eventually reaches a maximum. Notably, the maximum of $\langle M_{max}^{2nd} \rangle$ aligns with a curvature change in the $\langle M_{max} \rangle$ curve, furthermore, it coincides with the critical damage level $d_c$, previously identified as the position of the maximum of the average fragment mass $\langle M_{av} \rangle$. Below the critical damage $d_c$, the layer develops cracks; however, the presence of a dominant fragment, whose size remains comparable to the initial size of the layer, indicates that the system largely retains its structural integrity. This behaviour identifies the damage phase of the shrinking system. At the critical point, the formation of a connected crack network spanning the entire system results in the disintegration of the layer into a large number of fragments, each significantly smaller than the original size of the layer. In this fragmented phase further cracking gives rise to a gradual size reduction of fragments driven by the binary breakup mechanism.

To characterize the transition from the damage to the fragmentation phase of the shrinking layer, we carried out a finite-size scaling analysis. In particular, computer simulations were performed for four different system sizes, $R = 30, 60, 90, 120$, for each set of parameters considered. It can be observed in Fig. 7 at the adhesion strength $D_s/D_b = 0.08$ that the average mass of the largest fragment $\langle M_{max} \rangle$ normalized by the total mass $M_0$ decreases steeper around the critical damage $d_c$ in larger systems. The inset of the figure demonstrates that rescaling the data along the two axis using the scaling relation of the order parameter of continuous phase transitions [28, 36], the $\langle M_{max} \rangle$ curves obtained at different radii $R$ can be collapsed on the top of each other. The good quality collapse implies the validity of the scaling relation

$$\langle M_{max} \rangle (d, R) = R^{-\beta/\nu} \Phi \left( (d - d_c(\infty)) R^{1/\nu} \right), \tag{7}$$

where $\Phi(x)$ denotes the corresponding scaling function. In the scaling relation Eq. (7) the parameters $\nu$ and $\beta$ denote the critical exponents of the correlation length and order parameter, respectively, while $d_c(\infty)$ is the value of critical damage in the limit of infinite system sizes $R \to \infty$. The parameter values that provide the best collapse in Fig. 7 are $\nu = 1.33$,

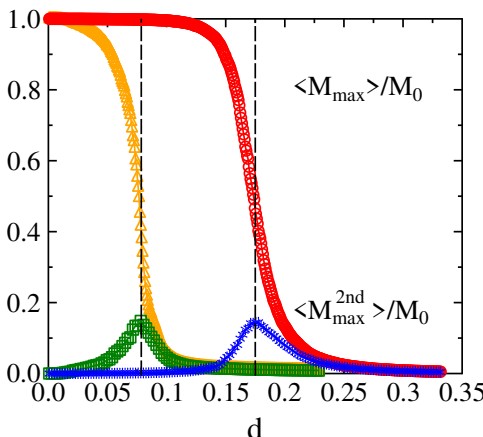

Figure 6: The average mass of the largest fragment $\langle M_{max}\rangle$ and the second-largest fragment $\langle M_{max}^{2nd}\rangle$ scaled by the total mass of the system $M_0$ for a system of size $R = 120$ at two different values of the adhesion strength $D_s/D_b = 1.6 \cdot 10^{-3}$ and $D_s/D_b = 0.08$ from left to right. The dashed vertical lines indicate the position of the maximum of $\langle M_{max}^{2nd}\rangle$, which coincides with the critical damage $d_c$ as determined from the analysis of the corresponding average fragment mass $\langle M_{av}\rangle$.

$\beta = 0.149$, and $d_c(\infty) = 0.15$. The result suggests that the damage-fragmentation transition of the shrinking layer shows analogy to continues phase transitions.

This analogy is further confirmed by the behaviour of the average fragment mass $\langle M_{av}\rangle$. Figure 8 shows that increasing the system size $R$ at a fixed adhesion strength $D_s/D_b$ the maximum of $\langle M_{av}\rangle$ gets higher and its position shifts to lower damage values. Hence, in layers of larger extension the connected crack network emerges at lower damage and larger fragments are created. The inset of the figure demonstrates that rescaling the data set using the finite size scaling relation of the susceptibility [28, 36], the $\langle M_{av}\rangle$ curves obtained at different $R$ values can be collapsed on the top of each other. The results indicate that the data obey the scaling relation

$$\langle M_{av}\rangle(d,R) = R^{\gamma/\nu}\Psi\left((d-d_c(\infty))R^{1/\nu}\right),\tag{8}$$

where $\Psi(x)$ denotes the scaling function [28, 36] and $\gamma$ is the critical exponent of the susceptibility. Best collapse was achieved with the exponent $\gamma = 2.4$, while for $\nu$, and $d_c(\infty)$ the same values were used as above.

For consistency, we also checked the validity of the size scaling of the critical damage

$$d_c(R) = d_c(\infty) + AR^{-1/\nu},\tag{9}$$

where the finite size critical point $d_c(R)$ of the system was identified as the position of the maximum of the $\langle M_{av}\rangle(d,R)$ curves at each $R$. Figure 9 shows that by plotting $d_c(R)-d_c(\infty)$ as a function of $R$, a power law is obtained with a reasonable quality using the same $d_c(\infty)$ value as above in Figs. 7 and 8.

At the critical point $d = d_c$ where the system fragments, and in the supercritical phase $d > d_c$, the snapshots in Figs. 3 and 4 reveal a well-defined average fragment size, indicating that the fragment mass distribution is not scale-free. Fragments emerge when a region of the layer gets completely surrounded by cracks, hence, the internal stress field of fragments evolves independently as shrinkage continues. This process leads to binary breakups, where a fragment splits into two smaller "daughter" pieces along a crack typically initiated near the fragment's center. Binary breakup dynamics have been shown to result in log-normal fragment mass distributions [31,37–39]. This is illustrated by Fig. 10, where the mass distributions $p(m)$

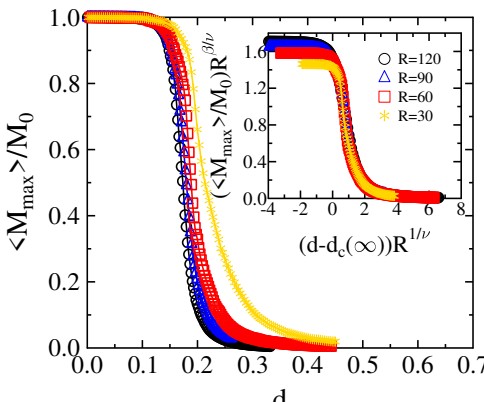

Figure 7: The average mass of the largest fragments $\langle M_{max} \rangle$ as a function of damage $d$ for various system sizes $R$ at the adhesion strength of $D_s/D_b = 0.08$. The inset shows the scaling plot of the data based on Eq. (7). The best data collapse is achieved with the parameter values $d_c(\infty) = 0.15$, $1/\nu = 0.75$, and $\beta/\nu = 0.112$.

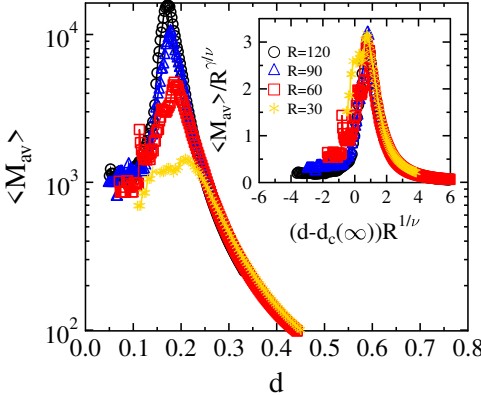

Figure 8: The average fragment mass $\langle M_{av} \rangle$ as a function of damage $d$ for various system sizes $R$ at the adhesion strength of $D_s/D_b = 0.08$. The inset shows the data collapse analysis of the curves based on Eq. (8). The optimal parameter values for the best collapse are $d_c(\infty) = 0.15$, $1/\nu = 0.75$, and $\gamma/\nu = 1.80$.

of the fragmentation phase $d > d_c$ rapidly attain a log-normal functional form

$$p(m) = \frac{1}{m\sigma\sqrt{2\pi}} \exp\left[-(\ln(m)-\mu)^2/2\sigma^2\right], \tag{10}$$

as the damage $d$ surpasses the critical value $d_c$. In Eq. (10) $\mu$ and $\sigma$ denote the average and standard deviation of the mass distribution. In the subcritical phase $d < d_c$, the fragment mass distribution exhibits two distinct regimes: a rapidly decaying function for small fragments and a pronounced bump corresponding to larger fragments, with a gap separating these regimes (see Fig. 10). As the critical point is approached from below, large fragments gradually break into smaller pieces, causing the bump to diminish and the gap to fill. Interestingly, the subcritical mass distributions display a power-law regime

$$p(m) \sim m^{-\tau}, \tag{11}$$

with an exponent $\tau \approx 2$. Note that the cutoff mass of the power law regime is practically the average mass of the second largest fragment $\langle M_{max}^{2nd} \rangle$, while the bump of the large fragments is formed around the average of the largest mass $\langle M_{max} \rangle$. With increasing damage,

the largest and second largest fragments become comparable at the critical damage $d_c$, hence, this is the point of the dynamics where the gap of the mass distribution disappears and $p(m)$ gradually transforms into a log-normal shape (see Fig. 10). It is important to note that Kooij et al. [14] also reported different functional forms of fragment statistics, i.e. exponential and power-law fragment size distributions in their sugar-glass experiments, where the two regimes were selected by the mode of driving (slow thermal vs. rapid impact). In our case, however, both distributions arise under identical slow loading conditions, and the selection is instead controlled by the degree of accumulated damage.

We repeated the finite-size scaling analysis at several values of the adhesion strength $D_s/D_b$ to examine how the range of load redistribution affects the universality class of the shrinkage-induced damage-to-fragmentation transition. The critical exponents were determined by optimizing the quality of the scaling collapse using the method of Bhattacharjee and Seno [40]: for a given pair of trial exponents in Eqs. (7, 8), the data for all system sizes $R$ were rescaled and compared by evaluating the average absolute deviation $P_b$ between each curve and the interpolation of every other curve, restricted to the range where their rescaled abscissae overlap. The optimal exponents were obtained by minimizing this deviation measure $P_b$, using a coarse grid search followed by a local refinement. Statistical uncertainties of the exponents were estimated by varying one exponent at a time until $P_b$ increased by 1% relative to its minimum, while keeping the other exponent fixed. This procedure was carried out independently in the positive and negative directions, yielding possibly asymmetric error bars when the $P_b$ surface was not symmetric around the minimum. Hence, the method provides an objective quantification of the quality of the scaling collapse and yields well-defined error estimates for the critical exponents [40]. As shown in Fig. 11, although the critical exponents $\beta$, $\gamma$, and $\nu$ have considerable uncertainties, their values fall reasonably close to those of the two-dimensional bond percolation model, $\beta = 5/36$, $\gamma = 43/18$, and $\nu = 4/3$ [28]. Larger deviations from the percolation values occur at the smallest adhesion strengths $D_s/D_b$, where the maximum crack extension becomes comparable to the system size (see also Fig. 5($b$)). In this limit, screening of the long-range elastic interactions becomes less effective as the system approaches criticality.

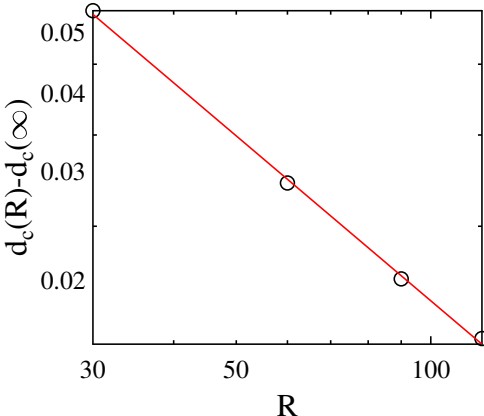

Figure 9: The difference between the critical damage $d_c(R)$ of finite-size systems and its asymptotic limit $d_c(\infty)$ for infinite systems as a function of $R$ at the adhesion strength of $D_s/D_b = 0.08$. The value $d_c(\infty) = 0.15$, which yields the most accurate straight line in the double logarithmic plot, is consistent with the value used in the data collapse analysis. The fitted power-law exponent is $1/\nu = 0.75$.

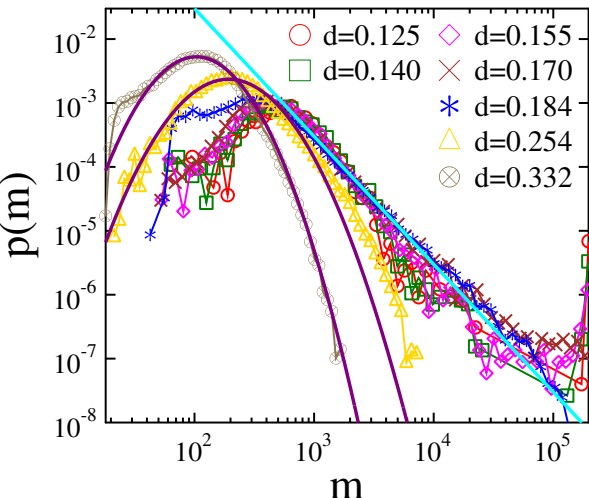

Figure 10: Mass distribution of fragments $p(m)$ for the system size $R = 120$ with the adhesion strength $D_s/D_b = 0.08$ at several damage values $d$ during the evolution of the system. The continuous lines represent fits with log-normal Eq. (10) and power law Eq. (11) functional forms. For this parameter set, the value of the critical damage is $d_c = 0.171$.

# 6 Discussion

Shrinkage-induced cracking of thin material layers attached to rigid substrates is a phenomenon widely observed in nature and holds significant potential for various industrial applications. This process typically evolves slowly, driven by either desiccation or cooling. In layers with heterogeneous material properties, numerous weak points act as nucleation sites for cracks, which grow in random directions, ensuring a high degree of isotropy. Over time, as a consequence of the accumulation and merging of cracks, the layer falls apart into a large number of pieces. The intricate crack patterns observed in phenomena such as drying lake beds are the final stages of this complex process. During the past decades experimental and theoretical studies uncovered the dynamics of crack nucleation and propagation in layers undergoing constrained shrinking, furthermore, the structural properties of final state fragmentation mosaics, and the geometrical features of fragments. However, the laws governing the evolution of the cracking layer as it shrinks remained an unsolved problem of fundamental importance.

In this paper, we carried out a theoretical investigation of the cracking process, focusing on the evolution of the fragment ensemble during shrinkage. Using a two-dimensional discrete element model, we simulated the breakup process under controlled conditions. The model discretizes the layer into a random lattice of space-filling convex polygons, capturing the quenched structural disorder of the material. Material elements, represented by polygons, are connected by elastic, breakable beams, while adhesion to the substrate is modeled with spring elements that exert a restoring force when displaced. Shrinkage is simulated by progressively reducing the natural length of the beams, introducing a uniform shrinkage strain in the layer. The time evolution of the system is followed by solving the equations of motion for the polygons removing overstressed beams iteratively. Simulations were performed at a fixed shrinking rate for four different system sizes, varying the stiffness ratio between adhesion springs and beams as the key control parameter. Our simulations revealed that the adhesion strength quantified by the stiffness ratio significantly affects the stress redistribution around cracks, shaping the stress field within the cracked layer.

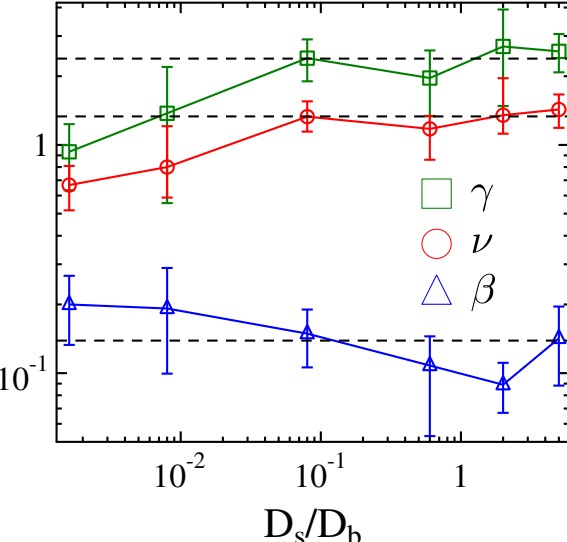

Figure 11: Critical exponents $\beta$, $\gamma$, and $\nu$ of the damage to fragmentation transition as function of the adhesion strength $D_s/D_b$. The dashed horizontal lines indicate the corresponding exponents of the two-dimensional bond percolation model [28].

Analyzing the fragment masses as shrinkage progresses, we identified two distinct phases of the evolution. In the *Damage Phase*, cracks develop within the layer but a dominant piece, significantly larger than the others, persists and maintains the structural integrity of the system. In the *Fragmentation Phase*, at sufficiently high damage levels, the layer breaks into numerous smaller fragments, all substantially smaller than the system size. The damage-to-fragmentation transition occurs at a critical damage value where the simultaneously growing cracks merge into a fully connected crack network spanning the system. Based on finite-size scaling of the simulated data, we demonstrated that at this critical point the shrinking system undergoes a continuous phase transition. To determine the critical exponents of the transition we performed a careful finite-size scaling analysis at several adhesion strengths, implementing the method of Bhattacharjee and Seno [40] to optimize the quality of scaling collapse.

A key finding of our study is that although the adhesion strength controls the degree of heterogeneity of the stress field in the cracked layer, it practically does not affect the universality class of the transition. In the limit of strong adhesion, stress redistribution around cracks is minimal, resulting in an almost homogeneous stress field. Consequently, crack growth is largely absent, and damage accumulation is dominated by the random nucleation of microcracks due to the material's structural disorder. In contrast, for weak adhesion, strong stress concentrations arise near crack tips, enabling cracks to grow and interact dynamically. Our analysis showed that the critical exponents fall reasonably close to those of the two-dimensional bond percolation model, indicating that shrinkage-induced breakup may belong to the percolation universality class [28]. In our system, the control parameter of the transition is the damage $d$, with a critical value $d_c$ at which the connected solid phase disappears. As an order parameter we use the relative mass $M_{\max}/M_0$ of the largest fragment. Near $d_c$, fluctuations of this scalar quantity dominate, while the elastic interactions governing crack formation are effectively short-ranged due to screening by the crack network. Moreover, the damage level at which a system-spanning crack network first forms coincides with the loss of mechanical connectivity. Larger deviations from the percolation values arise at the smallest adhesion strengths $D_s/D_b$, where the maximum crack extension becomes comparable to the system size and screening of long-range elastic interactions becomes less effective as criticality is approached.

While our critical exponents are close to those of the two-dimensional bond percolation model, we cannot exclude alternative universality classes given the limited size range of our finite-size scaling. Within our uncertainty, the exponents inferred from the scaling analysis are consistent with the percolation values, and the agreement improves as the increasing adhesion strength gives more controll for the disordered microstructure of the layer in cracking. We also verified that the optimal exponents remain stable under reasonable variations of the collapse window and interpolation scheme used in the Bhattacharjee–Seno method [40], reducing the risk that our conclusions hinge on fitting choices.

We further conjecture that introducing anisotropy into the mechanical properties of the layer—for instance, by applying an initial mechanical perturbation before desiccation sets in for a paste—could alter the universality class of the shrinkage-induced fragmentation transition [18, 25, 31].

Our simulations revealed that the fragment-mass distribution exhibits a characteristic evolution. Below the critical point, the distribution develops a power-law regime for small fragments together with a distinct hump for larger ones, separated by a gap. Above the critical point, fragments undergo a binary splitting process, which drives the distribution toward a log-normal form. It is instructive to compare these results with the experiments of Kooij et al. [14], who demonstrated that fragmentation of brittle sugar glass can proceed by qualitatively different mechanisms depending on the driving conditions. Under slow thermal loading they observed exponential fragment-size distributions, whereas rapid impact led to power-law statistics, even though the material and geometry were identical. These findings highlight that breakup processes may undergo transitions between distinct statistical regimes. Our system differs in that the driving is always slow, yet a transition from a damage-dominated regime to a fragmentation-dominated regime emerges as the accumulated damage approaches a critical value. Thus, while the control parameters differ (driving mode in [14] versus internal damage here), both studies indicate that breakup phenomena can exhibit sharp crossovers reminiscent of phase transitions.

In addition to the adhesion strength relative to the layer stiffness, the system has one further control parameter, namely the strain rate of loading, which sets the external driving. In our study, all simulations were carried out at a fixed strain rate in the quasi-static regime. In this regime, varying the strain rate primarily controls the time scale of evolution (e.g., the nucleation rate of microcracks) but does not alter the final crack topology or the fragment statistics [41]. We therefore do not expect the strain rate to influence the static critical behaviour reported here (critical point and exponents). Deviations may arise only at sufficiently high strain rates, where significant changes of the stress field around cracks can occur during avalanches of crack propagation.

We note that our discrete element model is constrained to two dimensions, which is of course a simplification compared to real three-dimensional fracture processes. However, many shrinkage-induced crack patterns of practical and geological relevance—such as desiccation cracks on surfaces, permafrost polygons, or columnar joints initiated at cooling surfaces—are effectively governed by two-dimensional crack networks at their interfaces. In this sense, the 2D model captures the essential statistical features of surface-constrained fragmentation, while providing the numerical tractability required for finite-size scaling analyses. Extending the present approach to fully three-dimensional systems would be a valuable direction for future work, allowing direct exploration of possible dimensional crossover effects on the universality class.

Our theoretical results embed slowly driven breakup phenomena of heterogeneous materials into the general framework of statistical physics. The scaling laws identified in this work are relevant both for the understanding of natural fragmentation patterns and for potential industrial applications.

# Acknowledgments

**Funding information** Project no. RRF-2.3.1-21-2022-00009, titled National Laboratory for Renewable Energy has been implemented with the support provided by the Recovery and Resilience Facility of the European Union within the framework of Programme Széchenyi Plan Plus. Supported by the University of Debrecen Program for Scientific Publication. Project no. TKP2021-NKTA-34 has been implemented with the support provided from the National Research, Development and Innovation Fund of Hungary, financed under the TKP2021-NKTA funding scheme.

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
