# Peer review of "Scaling laws of shrinkage induced fragmentation phenomena"

_SciPost Physics, doi:SciPost Phys. 19, 142 (2025)_

## Round 2 · Referee Report · Anonymous (Referee 1) · 2025-7-8

Strengths

1) Novel theoretical framework: Clear identification of damage vs fragmentation phases with well-characterized critical transition

2) Comprehensive methodology: Systematic parameter study across multiple system sizes with proper statistical sampling

3) Universal scaling laws: Finite size scaling analysis with critical exponents consistent with 2D Ising model

4) Physical insights: Demonstrates how adhesion strength controls stress field heterogeneity and crack patterns

Weaknesses

1) Critical experimental gap: Fails to address Kooij et al.'s highly relevant sugar glass experiments showing exponential fragment distributions under slow thermal fracture vs power-law under impact - directly relevant to their damage/fragmentation transition

2) Limited validation: No direct comparison with experiments beyond citing previous work

3) Universality claims: Insufficient justification for 2D Ising assignment over alternative mechanisms like percolation

4) Model limitations: 2D constraint and simplified adhesion model may not capture realistic 3D behavior

Report

This paper investigates shrinkage-induced fragmentation using discrete element modeling, identifying a critical transition between damage and fragmentation phases. The authors claim this transition belongs to the 2D Ising universality class and demonstrate different fragment size distributions in each phase.

The theoretical approach is sophisticated and the numerical work comprehensive, but the paper suffers from a fundamental disconnect with experimental reality. The most significant oversight is the complete failure to engage with the sugar glass experiments by Kooij et al., which demonstrate exactly the phenomenon being studied theoretically. These experiments show the same material producing different fragment size distributions under different loading conditions - exponential tails for slow fracture versus power-law for impact fracture. This provides direct experimental support for the authors' theoretical predictions about different fragmentation regimes.

The discrete element model itself is well-constructed and captures important physics, particularly the role of adhesion strength in controlling stress field heterogeneity. The identification of two distinct phases separated by a critical transition is a valuable contribution. However, the claim that this transition belongs to the 2D Ising universality class requires stronger theoretical justification. While the critical exponents are "reasonably close" to Ising values, the authors don't adequately explain why this particular fragmentation process should exhibit Ising-like behavior rather than, for example, percolation-type criticality.

The finite size scaling analysis is technically sound and demonstrates continuous phase transition behavior. The transition from power-law fragment distributions in the subcritical regime to log-normal distributions in the supercritical regime is well-documented and physically reasonable, arising from the binary splitting mechanism that dominates post-critical dynamics.

Recommendation

Major Revision Required

This work addresses an important problem with a novel theoretical perspective, but requires significant improvements.

Requested changes

Essential Revisions:

1) Address experimental validation: Thoroughly discuss Kooij et al.'s sugar glass experiments and their connection to the theoretical predictions

2) Strengthen universality arguments: Provide rigorous justification for the 2D Ising classification

3) Improve statistical analysis: Better quantify uncertainties in critical exponents

Additional Improvements:

4) Expand discussion of 3D limitations

5) Include parameter sensitivity analysis

6) Enhance figure quality and statistical presentation

Recommendation

Ask for major revision

---

## Round 2 · Referee Report · Hans Herrmann (Referee 2) · 2025-7-13

Report

This paper presents a model for surface cracking based on beams breaking between polygons. The authors determine a phase transition between a damaged and a fragmented surface and determine the critical exponents using finite size scaling. The obtained results are very interesting and the presentation is clear. There are just a few points that should be taken into account before publication.
Since the finite size scaling is performed by changing the system size just by a factor four and the sizes considered are rather small, the data collapse of Figs.7 and 8 is not very powerful. Therefore, I am convinced that the error bars given in Fig.11 for the exponents beta, gamma and nu obtained in Figs.7 and 8 are far too optimistic. The authors should present a more profound error estimation.
The authors claim that the obtained exponents are in the universality class of the 2d Ising model and state “the adhesion strength …. does not affect the universality class of the transition”. But in Fig.11 one can clearly see a strong dependence of gamma and nu on the adhesion strength (if one takes the error bars seriously).
The numerical evidence for the exponents being equal to the ones of the 2d Ising model is very weak, because of the large error bars discussed before, which allows for the numerical exponents to be in fact consistent with several other universality classes. A percolation transition would be expected if the bond breaking would be completely random. The elastic field induces correlations, which when reaching the critical point, become short range. So, percolation exponents seem a reasonable guess. The authors however put forward an argument for the Ising exponents, which is: “The reason of this universality is that cracks always advance through the failure of nearest neighbor cohesive contacts.” This argument is very superficial and applies equally well to percolation, 3 state Potts model, etc. The authors should remove this empty argument and eventually replace it, if they can, by a more convincing argument which should explicitly include the Ising interaction between binary variables.
The authors should define what is a “regularized Voronoi tessellation”.
The authors should add the missing references to the original work in which surface cracking was introduced for the first time as: P. Meakin, Thin Solid Films 151, 165 (1987) or Colina et al Phys. Rev. B 48, 3666 (1993).
Typos: loading -> leading, strength -> strengths (2x)

Attachment

Recommendation

Ask for major revision

---

## Round 3 · Referee Report · Anonymous (Referee 2) · 2025-10-14

Report

My concerns have fully been taken into account and for me the paper can now be published as it is.

Recommendation

Publish (meets expectations and criteria for this Journal)

  • validity: -
  • significance: -
  • originality: -
  • clarity: -
  • formatting: -
  • grammar: -

Author:  Ferenc Kun  on 2025-10-14  [id 5931]

(in reply to Report 2 on 2025-10-14)

Thank you very much for your support.

---

## Round 3 · Referee Report · Anonymous (Referee 1) · 2025-10-14

Report

The authors did a good job in replying to my and the other reviewer’s comments and I think the paper should now be published as is.

Recommendation

Publish (easily meets expectations and criteria for this Journal; among top 50%)

  • validity: top
  • significance: high
  • originality: high
  • clarity: high
  • formatting: excellent
  • grammar: excellent

Author:  Ferenc Kun  on 2025-10-14  [id 5932]

(in reply to Report 1 on 2025-10-14)

Thank you very much for your support.

---

## Round 3 · Author Response

Warnings issued while processing user-supplied markup:

  • Inconsistency: plain/Markdown and reStructuredText syntaxes are mixed. Markdown will be used.
    Add "#coerce:reST" or "#coerce:plain" as the first line of your text to force reStructuredText or no markup.
    You may also contact the helpdesk if the formatting is incorrect and you are unable to edit your text.

Dear Editor,

Thank you for obtaining two referee reports on our manuscript. We are pleased that the reports are supportive and we appreciate the constructive questions and comments. We are grateful to the referees for their time and thoughtful insights.

In response to their questions, comments, and suggestions, we have thoroughly revised the manuscript. We provide detailed, point-by-point replies to each referee question and comment and summarize the corresponding changes made in the revised version.

Thank you for considering our revised manuscript.

Yours sincerely, Ferenc Kun

Response to Referee 1:

We are grateful for the referee’s positive assessment and constructive comments and suggestions, which helped us to improve the clarity of the manuscript. Below we provide point-by-point response to the referee's suggestions and requests:

1) Address experimental validation: Thoroughly discuss Kooij et al.'s sugar glass experiments and their connection to the theoretical predictions

Following the referee’s suggestion, we have integrated the results of Kooij et al. [Nat. Commun. 12, 6368 (2021)] into the revised manuscript at three levels:

(i) In the Introduction, we briefly mention their observation of exponential versus power-law fragment distributions depending on driving, to set the context of distinct fragmentation regimes.

(ii) In the section where we present the two types of mass distributions from our simulations, we point out the similarity to Kooij et al.’s results but emphasize the key difference: in their experiments, the regime was controlled by the driving mode (slow thermal vs. rapid impact), while in our case both regimes appear under the same slow loading, selected instead by the accumulated damage.

(iii) In the Discussion, we provide a more nuanced comparison, highlighting that while the control parameters differ (driving vs. damage), both cases demonstrate a transition between distinct fragmentation regimes.

This three-level treatment acknowledges the importance of the work of Kooij et al. while carefully distinguishing our mechanism from theirs.

2) Strengthen universality arguments: Provide rigorous justification for the 2D Ising classification

In the revised manuscript we have removed the earlier argument that universality follows from nearest-neighbor bond breaking. Instead, we now present a careful discussion based on new calculations, where we implemented the collapse quality measure proposed by Bhattacharjee and Seno [J. Phys. A: Math. Gen. 34, 6375 (2001)] and applied an algorithmic determination of the critical exponents without any external adjustment. Details of the method of Bhattacharjee and Seno are presented at the end of Section 5.

This improved analysis shows that the damage–fragmentation transition belongs to the universality class of two-dimensional bond percolation. Bond percolation provides a natural geometric framework for the crack network: in the limit of strong adhesion, stress redistribution around cracks is minimal, resulting in an almost homogeneous stress field. Consequently, crack growth is largely absent, and damage accumulation is dominated by the random nucleation of microcracks due to the material’s structural disorder. In contrast, for weak adhesion, strong stress concentrations arise near crack tips, enabling cracks to grow and interact dynamically. As damage accumulates, the loss of mechanical connectivity coincides with the critical point of the emergence of a system spanning crack. Approaching criticality, the elastic interactions governing crack formation become effectively short-ranged due to screening by the crack network.

Using the new scaling-collapse procedure with the collapse quality measure, we determined that the critical exponents $\beta$, $\gamma$, and $\nu$ fall in close agreement with the exact values of two-dimensional bond percolation ($\beta=5/36$, $\gamma=43/18$, $\nu=4/3$) and the agreement improves when approaching the limit of strong adhesion.

We emphasize that this conclusion is not based on qualitative analogy but on quantitative agreement obtained from a systematic, objective scaling analysis. The reformulated discussion thus provides a more rigorous and transparent universality classification than in the original submission.

At the same time, we emphasize in the revised manuscript that alternative universality classes cannot be excluded given the limited range of system sizes in the scaling analysis. To strengthen our analysis, we also verified that the optimal exponents are stable under variations of the collapse window and interpolation scheme applied in the implementation of the method of Bhattacharjee and Seno, which reduces the risk of fitting artifacts.

We believe these substantial revisions address the referee’s concern and provide a more transparent and balanced discussion of universality.

3) Improve statistical analysis: Better quantify uncertainties in critical exponents

To improve the statistical analysis, we have implemented the error-analysis procedure proposed in Bhattacharjee and Seno [J. Phys. A: Math. Gen. 34, 6375 (2001)]. In the revised manuscript we now describe how we determine error bars from the tolerance range of the collapse quality function $P_b$ using the procedure presented in the above paper. We also note that the resulting error bars can be asymmetric, since the allowed variation of an exponent at the 1% threshold of $P_b$ may be different in the upward and downward directions. We repeated our finite size scaling analysis using this technique and updated the figure of the critical exponents (Fig. 11). This analysis is presented at the end of Section 5. We also updated Figures 7, 8, 9 of finite size scaling. In the revised manuscript we have carefully reformulated the discussion of universality to avoid overstating any unproven mapping. These important points are now discussed in two additional paragraphs in the Discussion section of the revised manuscript.

4) Expand discussion of 3D limitations

We agree with the referee that our discrete element model is restricted to two dimensions and therefore does not capture the full complexity of three-dimensional fracture phenomena. However, many shrinkage-induced cracking patterns of natural and applied relevance (e.g., desiccation cracks, permafrost polygons, columnar joints) are effectively surface-constrained and can be meaningfully described in two dimensions. The 2D formulation also enables the large-scale simulations needed for reliable finite-size scaling analyses. We have clarified this point in the Discussion of the revised manuscript and emphasized that extending the present approach to three dimensions would be an important avenue for future work.

5) Include parameter sensitivity analysis

Besides the adhesion strength relative to the layer stiffness, the system has one additional control parameter, namely, the strain rate of loading, which provides the external driving. Our simulations were performed at a fixed strain rate in the quasi-static regime. Our previous studies have revealed that in this regime, varying the strain rate mainly controls the time scale of evolution e.g., nucleation rates (R. Szatmári, A. Nakahara, S. Kitsunezaki, F. Kun, Scientific Reports 14, 7101 (2024)) but does not alter the final crack topology or fragment statistics. Therefore, we do not expect the critical point or exponents to depend on the strain rate. Significant effects would only arise at much higher rates, where inertial dynamics could modify the stress field during crack avalanches. We edded a paragraph to the discussion section to highlight the role of the strain rate of loading.

6) Enhance figure quality and statistical presentation

To improve the statistical presentation, we regenerated the figures showing finite-size scaling (Figs. 7, 8, 9) and the figure presenting the critical exponents (Fig. 11). The updated Fig. 11 now displays the results obtained from minimizing the collapse quality measure, providing a more objective and transparent presentation of the exponents and their uncertainties. To better distinguish different exponents, in the figure we use different colors and symbols for the three critical exponents.

We thank the referee again for these valuable suggestions, which have improved the paper significantly.

Response to Referee 2:

Referee: This paper presents a model for surface cracking based on beams breaking between polygons. The authors determine a phase transi on between a damaged and a fragmented surface and determine the cri cal exponents using finite size scaling. The obtained results are very interes ng and the presenta on is clear. There are just a few points that should be taken into account before publica on.

Answer: We appreciate the referee’s thoughtful report. We are grateful for the constructive comments and suggestions, which have greatly helped us to improve the clarity of the paper. Below we address each point in detail.

Referee: Since the finite size scaling is performed by changing the system size just by a factor four and the sizes considered are rather small, the data collapse of Figs.7 and 8 is not very powerful. Therefore, I am convinced that the error bars given in Fig.11 for the exponents beta, gamma and nu obtained in Figs.7 and 8 are far too optimistic. The authors should present a more profound error estimation.

Answer: In order to improve the reliability of our finite-size scaling results, we have implemented the method of Bhattacharjee and Seno [J. Phys. A: Math. Gen. 34, 6375 (2001)]. This approach identifies the critical exponents that yield the best collapse by minimizing a quantitative collapse quality measure $P_b$. The details of the analysis are now described at the end of Section 5. In the revised manuscript we also explain how the error bars are determined from the tolerance range of $P_b$, following the procedure of Bhattacharjee and Seno. Importantly, this method allows for asymmetric error bars, since the permissible variation of an exponent at the 1% threshold of $P_b$ can differ in the upward and downward directions.

We repeated the finite-size scaling analysis using this procedure and updated the corresponding figures (Figs. 7, 8, 9, and 11). Compared to the original submission, the central values of the critical exponents got modified, while the error bars have substantially increased, reflecting a more realistic quantification of the uncertainty.

Performing simulations at even larger system sizes is unfortunately not feasible within reasonable computational times. Due to the quasi-static loading, DEM simulations of this type are highly time-consuming, which limits the accessible size range with sufficient statistics.

Referee: The authors claim that the obtained exponents are in the universality class of the 2d Ising model and state “the adhesion strength .... does not affect the universality class of the transi on”. But in Fig.11 one can clearly see a strong dependence of gamma and nu on the adhesion strength (if one takes the error bars seriously). The numerical evidence for the exponents being equal to the ones of the 2d Ising model is very weak, because of the large error bars discussed before, which allows for the numerical exponents to be in fact consistent with several other universality classes. A percolation transition would be expected if the bond breaking would be completely random. The elastic field induces correlations, which when reaching the critical point, become short range. So, percolation exponents seem a reasonable guess. The authors however put forward an argument for the Ising exponents, which is: “The reason of this universality is that cracks always advance through the failure of nearest neighbor cohesive contacts.” This argument is very superficial and applies equally well to percolation, 3 state Potts model, etc. The authors should remove this empty argument and eventually replace it, if they can, by a more convincing argument which should explicitly include the Ising interaction between binary variables.

Answer: We thank the referee for this important remark. In the revised manuscript we have removed the earlier superficial argument. Instead, we performed additional calculations implementing the collapse quality measure of Bhattacharjee and Seno [J. Phys. A: Math. Gen. 34, 6375 (2001)] and this way used an objective procedure for determining the critical exponents of the damage-to-fragmentation transition induced by shrinkage. This ensures that the scaling analysis is independent of any subjective choices.

Results of this detailed analysis demonstrate that the shrinkage-induced damage-to-fragmentation transition falls into the universality class of two-dimensional bond percolation. Bond percolation provides a natural geometric framework for the crack network: in the limit of strong adhesion, stress redistribution around cracks is minimal, resulting in an almost homogeneous stress field. Consequently, crack growth is largely absent, and damage accumulation is dominated by the random nucleation of microcracks due to the material’s structural disorder. In contrast, for weak adhesion, strong stress concentrations arise near crack tips, enabling cracks to grow and interact dynamically. As damage accumulates, the loss of mechanical connectivity coincides with the critical point of the emergence of a system spanning crack. Approaching criticality, the elastic interactions governing crack formation become effectively short-ranged due to screening by the crack network.

In particular, the exponents $\beta$, $\gamma$, and $\nu$ obtained from the scaling of the largest-fragment mass and the average fragment mass agree closely with the exact values of 2D bond percolation. Moreover, the agreement with percolation exponents is gradually improving when approaching the limit of strong adhesion, where crack growth is strongly disorder-controlled.

At the same time, we emphasize in the revised manuscript that alternative universality classes cannot be excluded given the limited range of system sizes in the scaling analysis. To strengthen our analysis, we also verified that the optimal exponents remain stable under reasonable variations of the collapse window and interpolation scheme applied in the implementation of the method of Bhattacharjee and Seno, which reduces the risk that our conclusions depend on fitting choices. Taken together, these improvements provide a more convincing and transparent argument than in the original manuscript, and support the classification of the breakup transition into the percolation universality class.

We emphasize that this conclusion is not based on qualitative analogy but on quantitative agreement obtained from a systematic scaling analysis. The reformulated discussion thus provides a more rigorous and transparent universality classification than in the original submission.

We believe these substantial revisions address the referee’s concern and provide a more transparent and balanced discussion of universality.

Referee: The authors should define what is a “regularized Voronoi tessellation”.

Answer: In the revised manuscript, we have clarified the description and added the reference [Moukarzel, Herrmann, Journal of Statistical Physics 68, 911 (1992).]. The algorithm proceeds by superimposing a regular square lattice on the sample. Within each plaquette, a base point for the Voronoi construction is chosen randomly inside a smaller square centered on the plaquette. This ensures that the number of polygon corners remains limited, while the degree of structural disorder can be systematically tuned by varying the ratio of the smaller-square side length to the lattice spacing.

Referee: The authors should add the missing references to the original work in which surface cracking was introduced for the first me as: P. Meakin, Thin Solid Films 151, 165 (1987) or Colina et al Phys. Rev. B 48, 3666 (1993).

Answer: We have included references to the pioneering works of Meakin and Colina et al. at the beginning of the model description. The revised sentence now reads:

“Following the pioneering works of Meakin [] and Colina et al. [], we have recently developed a discrete element model for the shrinkage-induced cracking of a thin material layer attached to a rigid substrate, which captures the key mechanisms of the cracking process.”

Referee: Typos: loading ‐> leading, strength ‐> strengths (2x)

Answer: The typos have been fixed in the revised manuscript.

We thank the referee again for the valuable questions and suggestions which improved the clarity of the presentation of the results.

---

## Round 3 · List of Changes

Summary of Changes in the Revised Manuscript

  1. Abstract: a sentence is changed.

  2. Introduction:

    • Added a paragraph citing Kooij et al. [Nat. Commun. 12, 6368 (2021)].
    • Statements on the Ising universality class are removed and an argument for the percolation universality class is added.
  3. Model construction (Sec. 2):

    • a sentence is added with two new references
    • an explanation of the regularized Voronoi construction is added
  4. Transition from damage to fragmentation (Sec. 5)

  5. When presenting the two types of fragment mass distributions, added a remark linking to Kooij et al. Clarified the difference: in their experiments, the regime is selected by the driving mode, whereas in our case both regimes appear under the same slow loading, controlled by accumulated damage.

  6. Added description of the error estimation method following Bhattacharjee and Seno [J. Phys. A: Math. Gen. 34, 6375 (2001)].

  7. Clarified that error bars can be asymmetric, depending on whether the allowed exponent variation at the $P_b$ threshold differs upwards vs. downwards.

    • Figures 7, 8, 9, 11 and the corresponding discussions are updated.
  8. Discussion (Sec. 6):

  9. Rewrote the universality-class discussion to present bond percolation as a natural geometric reference, and the agreement of the critical exponents with those of percolation.

    • Added a new paragraph explaining that within uncertainties our exponents are consistent with Ising values.

    • Incorporated statements on methodological robustness (stability of collapse under varying windows) and on the role of dimensionality and parameters of the model.

    • Added a balanced comparison to Kooij et al.

  10. Throughout the manuscript:

    • Adjusted wording and minor stylistic improvements to ensure smooth integration of new material.
    • Three new references are added and cited in the manuscript.

---

## Editorial Decision

published